# An exploration of the emotional response among nurses in Bermuda, during the Covid-19 pandemic

Adam Moore[1‡]*, Navami Leena[2‡]

1 Independent researcher, Hamilton, Bermuda, 2 Department of Public Health, Coventry University, Coventry, England

‡ AM individual is the sole author of this work. NL individual is responsible for supervision during this work.
* writetoadammoore@gmail.com

**Data Availability Statement:** All relevant data are within the paper and its Supporting Information files

## Abstract

### Objectives

This paper utilizes an ontological approach to conduct a qualitative literature review in order to investigate the emotional impact of the Covid-19 pandemic on nurses internationally. After identifying common themes in the literature review, primary research is conducted to investigate the emotional impact of the Covid-19 pandemic on nurses working in Bermuda´s only acute healthcare facility.

### Methods

The literature review used the FRAMEWORK approach (Richie and Spencer, 1994, as cited in Hackett et al. (2018) to select a total of 16 papers for review, each of them qualitative primary research, aside from one paper reviewing international literature. Within the second part of this paper, investigating the experience of nurses in Bermuda, a grounded theory approach was utilised to collect primary data. Convenience sampling was used to recruit subjects to participate in one-on-one semi-structured interviews. Data saturation was achieved after 9 interviews. The FRAMEWORK method was utilised to analyse the interview transcripts, and identify, organise and collate themes.

### Results

The literature review highlights that nurses have strong emotional responses to caring for patients during the Covid-19 pandemic. Specific responses include: higher stress levels at work due to higher, more challenging workload, and a reliance on clinical leaders to assuage this stress. Stigma experienced outside of work lead to feelings of anxiety and isolation. Despite this, there was a common theme that nurses felt a satisfaction within their role to ´play their part´ in the pandemic. The primary research conducted in Bermuda demonstrates that there was an initial fear of Covid-19, receding as the pandemic developed. Much anxiety was due to a perceived lack of knowledge about the disease, lack of a coherent strategy, and inadequate equipment to protect staff, and properly care for patients. An increased level

**Funding:** The author received no specific funding for this work.

**Competing interests:** The authors have declared that no competing interests exists.

of cooperation amongst staff, and mutual support amongst nurses was noted, as was social stigma leading to feelings of isolation and anxiety. Social interactions and holistic activities were identified as primary resources to alleviate stress and uncertainty.

## Conclusions

Nurses in Bermuda elicited many of the same emotional responses as their international colleagues, as a result of caring for patients during the pandemic. These manifested as a result of higher, more challenging workload, and uncertainty about preparedness plan and quickly changing situations within the working environment. The results from this study can help formulate changes in policy to ensure smoother transitions to pandemic preparedness in the future.

## 1. Introduction

### 1.1 Effect of Covid-19 pandemic on nurses to date

The novel Coronavirus (Covid-19), first detected in Wuhan, China, in December 2019, has had a devastating impact worldwide. The World Health Organisation states that as of 28[th] April 2021, there have been 148,329,348 confirmed cases reported globally, including 3,128,962 deaths [1]. Nurses on the frontline of previous respiratory epidemics, such as Middle East Respiratory Syndrome (MERS) and Severe Acute Respiratory Syndrome (SARS), in 2002 and 2012, respectively, were under increased pressure due to challenging work environments coupled with staff shortages [2]. These factors, combined with fear of transmitting a disease contracted at work; compounded by stigmatisation in the community, lead nurses to experience occupational stress and anxiety, reportedly much higher than other healthcare disciplines [3]. Such emotional responses can lead to occupational burnout [4]. note that burnout and stressful work environment are the most commonly cited reasons for nurses considering leaving the profession. Research into the emotional responses of nurses working in the Covid-19 pandemic raises similar concerns [5].

The health and wellbeing of nurses is of paramount importance; and supporting nurses both practically and emotionally is essential to preserving their health, especially in the midst of a global pandemic [6]. Moreover, prior to the Covid-19 pandemic, the global nursing workforce was at a shortfall of 5.9 million nurses. This coupled with an aging workforce, struggling with increasing workloads and higher incidence of occupational burnout, is the predicted precursor to greater deficits in coming years. Pandemic-related burnout could therefore predict an exodus in experienced nurses [7, 8]. The safeguarding of nurses' guarantees to work in a safe, supportive environment where concerns are addressed; is essential in order to aid retention and recruitment of new staff.

The above literature highlights that nurses have strong emotional responses to caring for patients during the pandemic, for example, increase of stress and anxiety, which can have a long lasting effect, and can also cause nurses to leave the profession. At present, there is a lack of research into the experience of nurses working in Bermuda's response to the Covid-19 pandemic.

## 1.2 Impact of the virus in Bermuda

The global nursing shortage is felt in Bermuda; isolated as it is, in the Atlantic Ocean, with only one acute healthcare centre, King Edward Memorial Hospital (KEMH), serving the population of 62,000. The hospital's nursing staff quota exceeds the country's recently initiated nursing college training capabilities; and approximately 75% of nursing staff come from overseas; from a total of 25 countries (See S5 File in Supporting Information for Diversity Nursing Report). Bermuda saw its first cases on March 23$^{rd}$ 2020; with the initial spike in cases occurring in December 2020, peaking at 253 active cases; after which new infections declined- please see S3 File, below for a timeline [9]. Bermuda's isolated geographical location combined with limited bed capacity means that Covid-19 presents a risk to infrastructure being overwhelmed. Hospital Chief of Staff, Wesley Miller recognises that 'Our maximum capacity is not something that could be sustained for very long as our staff would be stretched. This type of burnout has been seen overseas' [10]. Demographically, Bermuda has aging population, with citizens over 60 anticipated to account for 24.9% of the population by 2026 [11]. Furthermore, with high density of patients living with chronic health problems make its residents particularly vulnerable to respiratory illnesses such as Covid-19 [11, 12].

## 1.3 Research questions: Aims and objectives

The study uses qualitative methodology to conduct an explorative study that aims to explore the emotional responses of nurses who treat Covid 19 patients. First, a literature review is conducted to better understand the global situation with the objective of gaining a better understanding of their experience. The researcher aims to answer the following questions with a view to gathering data that can used to inform policy to better support front line nurses during the current and future pandemics:

- What emotional responses did nurses have and what caused these responses?

- What problems did nurses face at work during the pandemic that lead to an adverse emotional response?

- How has life outside of work changed?

- What emotional response is associated with this change?

- What has generated positive emotional responses amongst nurses during the COVID 19 pandemic?

In the second part of this paper, primary research in conducted to answer these questions with regard to the experience of nurses working in Bermuda. Quality nursing care is essential to achieve good patient outcomes. With data from this study, it is hoped that interventions can be developed to support nurses in Bermuda to continue delivering compassionate, effective care; avoid burnout, and retain staff in employment during the ongoing-pandemic and thereafter.

## 2. Literature review

### 2.1 Nurses' emotional response to respiratory pandemics

The Covid-19 pandemic is an evolving situation, with researchers investing all resources to better understand, treat and eradicate the disease. Indeed, globally, of $421.3 million invested into research during 2020, only 1.8% was not specifically related to Covid-19 [13]. This demonstrates the urgency of the global situation. Nurses are at the forefront of any epidemic,

risking their lives to perform their duties. During the course of their work, they are likely to encounter traumatic experiences which elicit strong emotional responses, although not everyone experiences the same degree of emotional impact [7, 14].

Pandemics cause exhaustion, physical discomfort and strain on nurses, threatening their health, well-being and ability to perform their jobs. Challenging situations experienced within nursing work can lead to ´secondary PTSD, compassion fatigue and vicarious traumatization´ which are used almost interchangeably to describe the 'cost of caring' during the line of work [15, 16]. Further physical and emotional strain on an already stressed workforce elicits strong emotional response. Evidence shows nurses working in respiratory epidemics are likely to become stressed at work; and suffer knock-on effects in home life; due to difficult conditions at work; stigma experienced in the community, and isolation due to self-quarantine due to concern about passing the disease to family members [17]. These emotional responses are complex and multi-faceted, and cannot easily be defined with tick boxes and scales [18].

With the aim of achieving an understanding of what emotional responses respiratory pandemics have elicited from nurses in the past, the researcher carried out initial searches on CINAHL, MEDLINE and Google Scholar into nurse experiences during the SARS and MERS epidemics.

## 2.2 Search strategy

For the purposes of this initial literature review, initial searches were conducted on 1st March 2021. Boolean operators used:

Nurse AND attitudes AND COVID 19

Results were limited to full-text, academic Journals, published 2019–2021, in English language. Utilising CINAHL yielded 103 matches, the majority of which had no link to full text, with many results not focused on nurse experience. MEDLINE, yielded 39 results after exclusion criteria was applied; but again, full-text articles were hard to locate, and the majority were not relevant to the subject of interest. After consulting with Brunel Library staff, the researcher used the same Boolean operators on Scopus. Please see (S1 File) PRISMA chart of Scopus search strategy, below.

## 2.3 Literature review: Research methods and analysis

The FRAMEWORK approach was used to assess the quality of results; as it is useful to interpret a dataset with particular research questions in mind [19], in [20]. Of the 16 studies chosen for the literature review (see Appendix A in S1 Appendix), 6 were large-scale primary research, utilising online clinical questionnaires to assess respondents' views or emotions. The use of scales such as Depression, Anxiety, and Stress Scale-21 (DASS-21);and the Maslach Burnout Inventory, enable researchers to collate and compare results from one population to another, examining the impact of events or interventions, as numerical scores mean that like can be compared for like. Using online questionnaires with Likert scales ensures a much higher response rate than arranging interviews, for example, which require more time and resources.

Aside from one international literature review [21] all other papers reviewed primary data, using semi-structured interviews, allowing for more in-depth, personalised answers to questions, and freedom to discuss the matters that are important to participants. While conducting interviews are a useful way to gain an insight into an individuals' point of view; data is not transferable to another population, or even another individual working within the same healthcare facility. Results from these qualitative studies give personal, unique and detailed insight into the participants' emotional response; but are not generalizable; however, similar

responses being given by participants within the same facility can give a picture of broad findings within a group.

Wuhan's status as the epicentre of Covid-19 mean that research into nurses' experience in China is over-represented in the sample; with a total of 9 out of 16 studies analysed conducted in the country. Research conducted in Wuhan by [22–24] all illuminate the emotional response of increased stress as a result of higher, challenging workloads for nurses during the initial outbreak. Furthermore, [25] applying existence, relatedness and growth need theory to this group, recognising the importance of interpersonal relationships for nurses working with Covid-19 patients. [26] establishes that emotional response of nurses in Wuhan can be shaped by effective leadership; inclusive leadership styles helping to sooth stress and diminish work-based anxiety.

International research from every other continent triangulates findings from China. [27] found that nurses in Turkey experienced increase in anxiety; with [28] presenting supporting data from Paris, France and Andalusia, Spain, respectively. While some studies were focussed on a particular aspect of the nursing role, shift patterns, for example; all data reinforced key themes: nurses' emotional response of stress to higher workload within the workplace; stress due to stigma outside of work, perceived or otherwise; and a positive emotional response of satisfaction at doing job well, and feeling of dedication to the profession despite tough times [29, 30]. The FRAMEWORK analysis model [17], as cited in [18] was used to identify themes within the studies selected. Please see Fig 1 for a pictorial representation of themes identified. These themes are discussed in full below.

## 2.4 Discussion of themes

**2.4.1 Stress due to challenges at work.** All studies found that increased workload as a result of the Covid-19 pandemic lead to increase in stress, and increase in chance of experiencing anxiety and depression. The potential for healthcare workers to suffer from longer term, Post-Traumatic Stress Disorder (PTSD) type illness is identified; and it is established that PTSD is more prevalent among regular staff rather than non-acute or agency nurses; while front-line worker are most likely to suffer burnout [31]. Nurses discuss the challenges of higher workload meaning that they are not always able to deliver the care they would like to; due to shortages of both time and resources. This 'erosion of care' is seen as affront to their dedication to the role [30]; While each paper reviewed represents nurses as both dedicated to the professional and increasingly stressed; the urgency of work-based support [32] and wider state-based policy to ensure nurses needs are met is highlighted frequently from global data [21].

**2.4.2 Satisfaction in nursing role.** Nurses are proud to be working to look after Covid-19 patients, and proud to be an integral part of the pandemic response; this is a positive emotional response to increased workload. Nurses, in general, take pride in their work, and having a sense of duty to work during time of international uncertain [26]. In Wuhan, there is a willingness to care for Covid-19 patients; who grew when equipped with pandemic-related training and guidance [24]. Nurses are consistently shown to be motivated to engage with Covid-19 patients in acute environments; but need support, both in work, with role-specific training and education, and also out-of-work, with time to rest, relax, and spend time with loved ones [33].

**2.4.3 Stigma driven isolation and anxiety.** The stigma of being a nurse during the pandemic elicited strong emotional responses, leading to further stress and therefore a reduction in quality of life [25]. Each one of the 16 studies reports that nurses feel that they cannot socialise; and cannot see friends and family- due to self-regulation because of concern about being vectors; and perceived suspicion of transmitting disease contracted at work to others in the community. There is recognition that it is a human need to spend time with loved ones; and

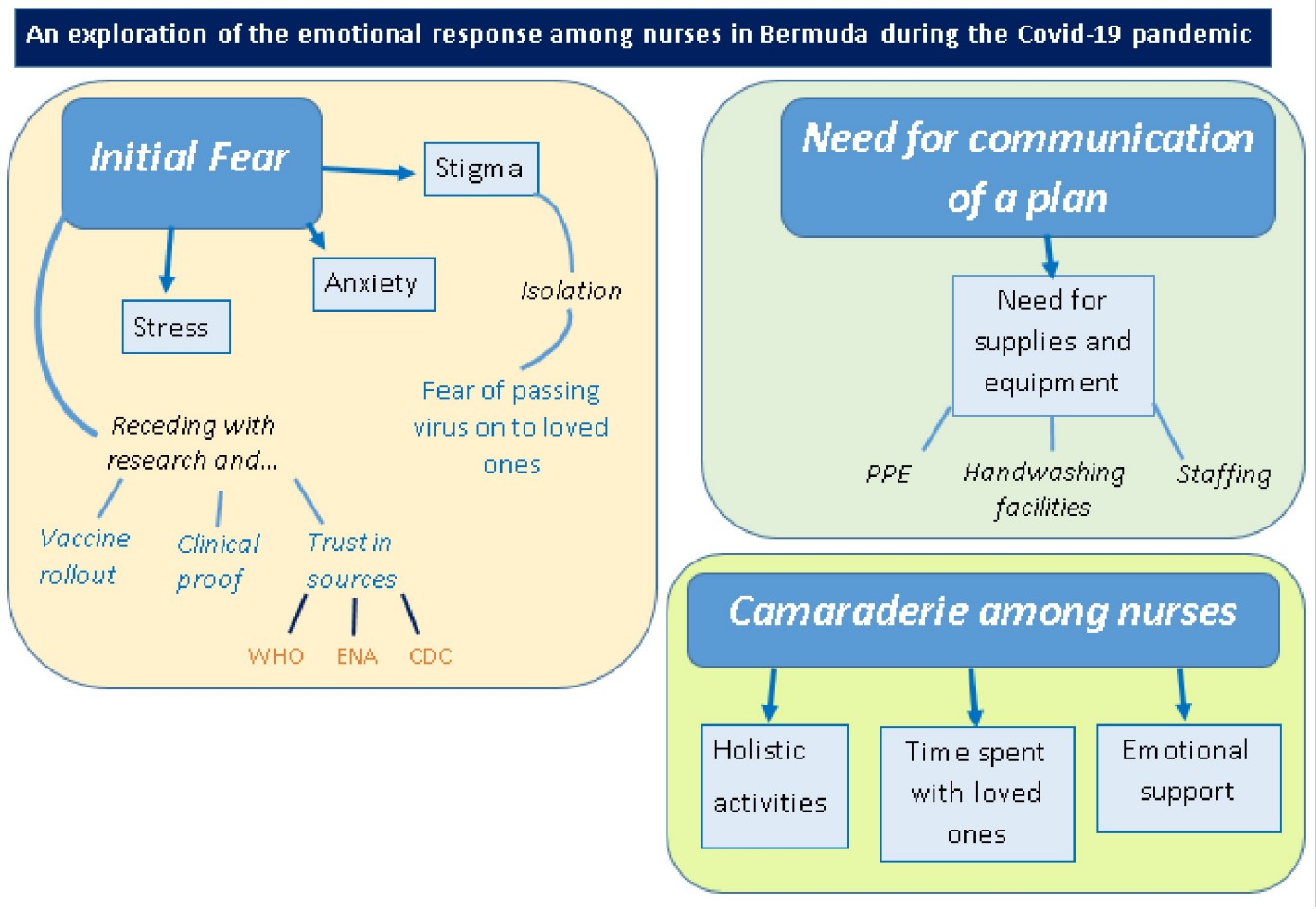

**Fig 1. Pictorial representation of themes identified.**

understanding that family and private life is connected to workplace life [22]. New nurses are identified as being in need of particular support, and need for all nurses to have training, support and counselling [34]. These interventions are identified as being the responsibility of national governments and policy makers, but also healthcare facility managers and clinical nurse managers. Academics highlight the need for more research in order to inform policy that better protects nurses [8].

**2.4.4 Reliance on clinical leaders to assuage stress.** A common thread through the literature was the impact that those within leadership roles could have on the emotional state of others working within the healthcare organisation. Managers with inclusive attitudes were found to quell stress and anxiety by having clear communication with nurses [35]. Management teams within the hospital are identified as being in charge of the practical management of patient flow- i.e. addressing bed capacity issues or staff shortages; but also serve as a visible and accessible source of practical and emotional support during stressful times [36]. Managers set the tone of the organisation, and in doing so can alleviate stress and anxiety amongst team members; with an inclusive leadership style facilitating clear communication, both top-down and bottom-up. [37, 38] shows that nurses internationally value managers who are responsive to nurse needs, in terms of practical issues- i.e. access to personal protective equipment (PPE), and also emotional support- having a debrief after stressful events, for example.

## 3. Research design and methodology

### 3.1 Methodology

The ontological approach utilised in the above literature review reveals that qualitative data yields the deepest, most thorough understanding of thoughts and feelings of an emotional nature. Therefore, focussed, semi-structured, one-on-one interviews were chosen in order to gain a deep understanding of the emotional responses of nurses.

The overall study design uses a grounded theory approach to data collection and analysis. Please see (S2 File) Gantt chart below (3.1.1) for full chronology of planning,, conducting and analysing data. Structured one-to-one interviews using open ended questioning were conducted, via Zoom. All interviews were carried out between 5th April and 12th April 2021. For interview outline please see Appendix B in S1 Appendix. Interviews were recorded with the Otter transcription app; reviewed manually for accuracy; and analysed using the FRAMEWORK method [17], as cited in [18]. Data was stored on a hard drive and kept locked in the researchers' home with all files password protected.

### 3.2 Sample and recruitment

The sample consisted of nurses working at KEMH, Bermuda. The advertisement poster (See S4 File in Supporting Information) states that the researcher is interested in attitudes of nurses working with Covid19 patients. The poster was displayed in staff rooms within Acute Departments within the hospital; as well outside the Staff Canteen, and on the hospital intranet.

During the interview process, positive cases were on increasing in Bermuda (See S3 File, section 1.2). Motivated by the impetus to find willing participants who would find time between busy shifts, convenience sampling was chosen to recruit subjects. Nurses interested were to respond to an email address, and provided with a Participant Information Sheet (PIS) and Consent Form (Appendices C) Please see Table 1, below, for full sample information. After PIS were reviewed, and written consent had been given, email was used to arrange a mutually convenient time to meet for a Zoom interview.

The researcher initially planned to collect data from 16 participants; as this figure meets the aim of research to be small enough to make the data manageable, yet large enough to provide a rich, textured understanding of the experience [39]. Time restraints, however, encouraged the researcher to limit numbers further. It was found that questions had been fully answered and the natural conclusion of the interview was usually around 15 minutes in length, although some were up to 25 minutes. Thematic saturation had been reached at 9 interviews; in that no new ideas or viewpoints were being propagated; Interviews were terminated at this point.

### 3.3 Ethical issues

It is of paramount importance to uphold the highest professional and ethical standards within research, and clinical practice, in order continue to improve patient care [40]. With this in mind, all participants were fully informed about the objectives and design of the study after expressing their interest. The anonymity of participants; and confidentiality of their responses, was ensured at every step of the research process. Data was protected to preserve the identity of those interviewed; as well as individual characteristics of any other patients or staff mentioned within interviews. The opportunity was given to ask questions, and while some of the participants were known professionally to the researcher, no participants were put under any pressure to take part. Participants were reminded that they could withdraw at any time. Interviews were anonymised, without names assigned, and the audio recordings and transcriptions were treated with the highest confidence.

**Table 1. Distribution of sampling among nurse subjects.**

| Participant number | Gender | Years of Nursing | Years at BHB | Achievement | Number of Covid patients nursed (approx.) | Country of training |
|---|---|---|---|---|---|---|
| 1 | M | 10 | 8 | Masters | 50 | Philippines |
| 2 | F | 11 | 4 | Masters | 30 | St. Lucia |
| 3 | F | 11 | 3 | Bachelors | 10 | Ireland |
| 4 | F | 42 | 32 | Bachelors | 12 | England |
| 5 | F | 13 | 3 | Bachelors | 20 | Dominica |
| 6 | M | 7 | 0.5 | Bachelors | 300 | England |
| 7 | F | 8 | 4 | Bachelors | 10 | Ireland |
| 8 | F | 12 | 4 | Bachelors | 5 | Jamaica |
| 9 | F | 3 | 3 | Diploma | 10 | Bermuda |

The particularly sensitive; emotive nature of the matter at hand means it is imperative that careful consideration is applied to participant aftercare. The researcher used clear and compassionate communication style in all contact with participants, from initial recruitment, the interview process, aftercare, and communication of results. Utilising a friendly, professional demeanour encourages knowledge sharing, recognising the value of participant contributions, as nurses and human beings [41, 42]. Further to these aims, and in keeping with local and international research standards, the research proposal was reviewed and approved by Brunel Research Ethics Online; by the Bermuda Hospitals Board, and also the Bermuda Ethics.

The most accurate method of establishing reliability and validity within qualitative research is the subject of much debate. [43] suggests that trustworthiness within qualitative data can be achieved by analyzing data with accepted analytical techniques and methodologies that can be assessed for credibility. The researcher has made every attempt to do so by way of transparent presentation of participant recruitment, data collection, transcription and coding, as well as the use of the FRAMEWORK analysis tool [17], as cited in [18]. Additionally, [44] notes that researcher bias can be a particular threat to validity in a study such as this, with an exploratory nature. The researcher makes every effort to be aware of his own biases, preconceptions and assumptions, exercising reflexivity throughout the entire data collection and analysis process.

## 4. Findings

### 4.1 Themes identified as an outcome of the analysis of data

**4.1.1 Initial fear, receding as pandemic develops.** All nurses interviewed expressed that the initial emotional response at the outset of the first wave of Covid-19, was one of fear. This caused stress and anxiety, mostly due to unknown nature of symptom effect and transmission. Words such as 'frightened', 'afraid', 'wary', 'nervous' and 'fearful' were used in relation to initial emotional response; and very much intertwined with feelings of stress and anxiety. Participant 5 (P5) succinctly explains how these emotional responses are inter-related:

*'Initially there was a lot of fear. We weren't clear on what PPE we were supposed to be using. . . I felt afraid because it felt like, you know, you go into work and you might catch Covid and die.' (P5)*

A recurring theme from participants experiencing fear and anxiety, was concern about transmitting the disease to those close to them, leading to self-imposed exclusion from loved ones, often family residing in the same home. This brought about emotional response of loneliness and feeling of isolation; explored later in the 'Social stigma' section. Initial emotional

responses of fear and anxiety are quelled by better clinical understanding of how the disease is contracted, and treated, which gave nurses hope, and also provided peace of mind while treating Covid-19 patients. P9 states:

'*Its brand new, so of course you're going to have a level of anxiety. . . after understanding a bit more about it, you realise okay, let me do when I needed to do to protect myself, so I will go in and ensure I wear the proper PPE.*'

The majority of participants stated their fear, anxiety and stress levels were assuaged by the development and steady roll out of the Covid-19 vaccination programme. P4 discusses emotional response to the third spike in cases, when many staff has been fully vaccinated; explaining that while nurses might feel reassured to be vaccinated; the nature of the disease remains an unpredictable and challenging one to treat:

'*I feel like we feel a bit better because were vaccinated, like the staff seem to be a little more confident, but you still have that anxiety . . .because they're in a room and you can't pop in and out to assess them. So you're looking through a window at them all the time. . . that still brings up kind of fear, for the patient for the most part this time.*'

**4.1.2 Anxiety due to lack of knowledge and perceived lack of equipment.** All participants' emotional response to initial lack of information about the disease was anxiety and panic. Elevated stress levels were exacerbated by debate about what constitutes appropriate PPE; and also that supplies were insufficient for nurses to carry out their job roles safely. P9 states

'*In the beginning I think I was one of those people who were wary and kind of. . . unsure. . . because there was so little research on what proper PPE is and how [the virus] is transmitted. It was just kind of, the unknown, I had a bit of. . . not anxiety, but just unease in caring for the patients. Now that we know a bit more about it, and how it's transmitted, and the proper PPE for it, it's just like any other patient.*'

Participants universally felt less anxious, and more confident in caring for patients at work, and therefore less stressed outside of work, when they felt they had the correct PPE to protect themselves. Provision of this equipment is regarded as the job of hospital bosses.

Several participants voiced frustration that they were not provided with hand washing facilities outside of patient rooms; meaning they were unable to follow international guidelines about hand hygiene during donning and doffing PPE. P9 states

'*you're being told to do one thing, and it's not being able to be facilitated through the lack of a sink outside of the room*'.

P8 had similar concerns:

'*I'm already taking all this stuff off in the room. . . and when I come outside, I don't only need, I don't only want to sanitise. I want to wash my hands. That's all*'.

Whereas other staff felt their needs were met, and that lessons had been learnt during the fluctuation of cases. An example is provided by P1:

'*We asked for better protective gowns. . . we had this meeting and the manager was involved, HR was involved. . . because, you know, since we are the Covid floor and we have demands, he had to make some changes. . . like better PAPRs [Powered Air Purifying Respirators]. . . we demanded it and we got it*'.

Participants from this department felt encouraged and supported that their concerns were listened to; and that staffing levels were improved on the ward. The addition of tables outside patient rooms that could be used as stations to safely don and doff PPE meant staff felt more comfortable and confident in following guidelines for protecting themselves at work, and in doing so, assuaging emotional response of stress and anxiety about transmitting the disease to their close contacts at home.

**4.1.3 Anxiety due to perceived lack of plan.** P7 states there is conflict between advice given by clinical managers and shift supervisors on the appropriate PPE and equipment; adding to the anxiety felt by staff that they are not properly protected,

'*Then other departments. . . all wearing N95s, ICU are all wearing N95s, fully dressed. So just huge conflict*'.

P3 also voices emotional response of concern and frustration at conflict between messages on correct protection:

'*Last night we had house cleaners on the floor cleaning rooms in hazmat suits. And we're just making do with tiny little masks and the odd shield when we can get them. Really, I don't know.., it's just like nurses are expected to carry on and manage, it doesn't matter what the situation is.*'

Participants also felt there was a lack of communication about preparedness for increasing Covid-19 numbers, particularly before each spike, eliciting emotional response of concern and frustration. Staff interviewed used words such as 'abandoned' (P3) about a lack of communication about the practicalities of bed capacity and management, and perceived lack of 'a plan'. P3 states

'*They may well have a plan but they weren't voicing it to us, you know so you're kind of wondering what are they going to do next–you're worried, because you are the one that's going to deal with this, you know if there's a delay in somebody getting an ICU bed, it's you that has to look after the patient until that point. So, they need to be there to reassure us that there are doing something. . .. As soon as the Covid comes, you don't see management as much.*'.

Clinical managers and shift supervisors in charge of the hospital at nights and weekends are the link between top level management and floor level staff; and as such, are relied upon to provide support when staff need it. The role of managers and supervisors is not only practical in terms of supplies and staffing; (although these both have an impact on the stress levels of nurses at work) staff also relied on managers for reassurance. P5 discusses a conversation with her clinical manager after a week of surge in positive patients:

'*I spoke to my manager, and said to her 'What is the plan? Because I thought, you know the numbers are going up . . . She said we don't really have a plan, you know, we just do the same thing that we've been doing. . . and really, I would have preferred if we had an optimum plan. . . when it gets to the point that we can't take anybody else in, what do we do?*'

**4.1.4 Social stigma leads to a feeling of isolation.** All participants disclosed that their status as healthcare workers (HCW) made them feel stigmatised within the community, eliciting an emotional response of feelings of isolation and loneliness. While nurses interviewed explained that they had self-imposed isolation measures, out of fear of transmission to friends and family; they also reported decrease in socialisation due to perceived stigma of members of the public, explains P2:

'*I really didn't want to go anywhere, or be seen to be going anywhere, because you're a nurse and you're in the hospital and you don't want to talk to anybody else, especially those that aren't working the hospital, because you just don't. You're afraid you're going to give it to them or something.*'

This stigma was experienced among friends; P6 states:

'*Even recently I went for a meal with non-nurse friends. . .and it's like. . . you kind of feel, you know, they. . .I swear they treat you differently. Because they're worried as well, aren't they?*'

Also in public, with next-door neighbours; P7 speaking about an interaction with their landlady, who lives next door:

'*I parked my bike, and then I was waiting to speak to her, and she wouldn't put down the window. . . and then I realised that she was waiting for me to get off my bike, so she could get out of her car and go into the house.*'

**4.1.5 Increased cooperation at work; mutual support amongst nurses.** All participants had the emotional responses of confidence, courage and felt a morale boost from the mutual support and cooperation between their nursing colleagues. Most nurses reported that their first port of call during troubled times were their colleagues, providing both practical and emotional support when things are difficult, and enjoying a 'shared understanding' of the mentally challenging aspects of the job. P1, working on the initial Covid-19 cohort ward stated

'*Because of the pandemic we became, our floor became more of someone, we were united because we formed. . . a group. . .and everyone was just so cooperative and it made us unite. . .. So I think yeah, having the co-workers support at that time really helped us.*'

The closeness exhibited among colleagues, and nurses' reliance on their co-workers for emotional support, as well as practical matters, is a response to the high workload and often upsetting events in working life of acute care nurses in a pandemic:

'*We support each other. . . we have a group chat where we check up on one another and see how we're doing. You know if we have, if we had a bad shift or a busy shift, you know, we kind of like debrief and speak to one another. Just basically support each other, you know somebody afterwards, somebody might call and they might see you cry, you know, you might just need the moments of encouragement, and let them know that they're not in it alone and you're there as well.*'

All interviewees expressed gratitude for the emotional support provided by their colleagues during difficult times- the mutual understanding enjoyed by co-workers was repeatedly highlighted as an important factor to the closeness of these relationships. P6, when asked if any changes could be made to support nurses, stated

'*People are talking about how [the pandemic] actually affects peoples mental health, the actual things that are stressful. . .nothing gets done. Nothing. It's like, the black sheep if it, nothing gets spoken about. We need more communication*'.

Whereas P5 felt her feelings were addressed, stating that as numbers rose during each wave,

'*I think our managers surely ask us, do you all think you're ok? Do you feel so stressed you need to speak to somebody that can help*?'

Notably, this nurse was the only respondent that expressed knowledge of the Employee Assistance Programme (EAP) counselling service.

**4.1.6 Social interactions and holistic activities outside of work.** As discussed, nurses interviewed experienced a level of loneliness and isolation due to their status as healthcare workers. The most oft-cited resource to counter emotional response of these feelings was interactions with loved ones; closely followed by holistic activities- time spent outside; enjoying nature, or exercising. When asked what they could do to make themselves feel better outside of work, P7 state

'*Spinning. Exercise, basically. Yeah, 'cos when they gym shut down I had a mental breakdown. . . that's kind of the only thing; and I picked up drawing again, and I'll be doing my gardening. Lonely activities, basically.*'

P9, meanwhile, also expresses that exercise helps calm the stress of emotional response associated with a busy or stressful shift:

'*Being active, just getting out there and just doing things makes you feel better. . .Because you know, you get so caught up being tired, and you go home, eat and sleep, and then you just come back to work, and it's kind of depressing.*'

P2 speaks of their partner, stating that

'*We have curfews and lockdowns, I have company. . . so that made a difference. . ., we cook stuff, we go outdoors and go for walks, enjoy nature, pray.*'

P3 echoes this sentiment,

'*You know just go. . . forcing yourself to get up and go for a walk or just clear your head a bit. I haven't taken up yoga or meditation . . . Just having, you know, a good few close colleagues were able to talk it through, definitely helps me, you know*'.

Participants with live-in partners valued them highly as a resource for emotional support during times of high stress and anxiety, although several nurses disclosed that it helps them to speak with someone who '*has an understanding and appreciation of what [we]* do' (P3). The

majority of participants report that they rely heavily on their colleagues for support outside of work as well as inside; even prior to Covid-19. P5 states that

'*Before covid, I'm not much of a social person. So most of my friends are people who work in [my department] so we hung out together. . . yeah, we support each other.*'

While not being able to see friends and family face-to-face; nurses valued the knowledge that their loved ones are healthy. P8 states:

'*I think probably for me, just knowing my people are safe. My friends, my coworkers, my family. Everyone is ok, and Covid-free. That is comforting.*'

This confidence in health stems in part from the continued global rollout of the vaccine programme. The majority of participants (7/9) stated that their stress and anxiety levels were reduced and they felt more confident at work; and more hopeful for the future both inside and outside of work, due to the vaccine. P4 expressed concern that

'*We have a very large nursing population that won't take it, but they are seeing the results of Covid first hand. . . that bothers me a lot. How do we convince the public?*'

## 5. Discussion

The FRAMEWORK analysis tool [17], as cited in [18] was used to identify, organise and collate the themes discussed above. It was found that the emotional responses of nurses in Bermuda were reflected in the literature from elsewhere, globally. Moderate to severe stress, anxiety and depression symptoms in the early stages of the pandemic were found in a study of more than 3,000 nurses in New Zealand and Australia as evidenced in [45]. While this Bermuda based research does in no way aim to diagnose mental health conditions; the phrases used by participants, 'anxiety', 'stress' and 'panic', illustrate how they were feeling during the early stages of the pandemic. The concern voiced by P5 that they might come to work 'and die' is a reflection of global events with regards to HCW dying as a result of workplace exposure; as of May 2020, a total of 152,888 infections and 1413 deaths were reported amongst healthcare workers worldwide [46].

Participants are anxious to ensure that they are able to meet international standards on PPE and hand hygiene; in order to care for patients effectively; and also keep themselves and their families safe. This concern is shared with nursing colleagues in Guangdon Province, China, during the initial outbreak [6] Health promotion, and the provision of practical supplies essential for treatment delivery, is the responsibility of national governments and international partnerships. The Covid-19 pandemic brought about a multiagency approach to service delivery; with partnerships outside of the healthcare sphere; for example, UK-based household appliance manufacturer Dyson producing ventilators for use in hospitals [47]. Bermuda, small and isolated as it is, takes some cues from UK healthcare guidelines, and some from USA, resulting in compounding confusion among nurses concerning PPE. A global shortage, and debate concerning best practice with regards to PPE, i.e. utility of surgical masks vs. N95 respirators [48]; sets the backdrop for anxieties concerning availability of protection to avoid unnecessary risk at work. In March 2020 the WHO note that stock shortage is putting nurses at risk; leading planners to question whether it could be advisable to reuse or customize what had been considered one-use-only equipment [49]. Intersectionality within the procurement and allocation

of essential supplies- PPE; testing kits, and later vaccines, meant that the UK was able to provide Bermuda with much needed supplies [50].

Advances in research, coupled with the utility of newly developed vaccines; means that nurses in Bermuda feel better protected against the virus, and therefore less anxious at work; and ergo, in homelife. The reliability of vaccination is cited by the majority of nurses as important to assuaging stress around contracting Covid-19. 2 of the 9 nurses interviewed disclosed that they had chosen not to be vaccinated as time of data collection; demonstrating that healthcare professionals are subject to the same anxieties concerning vaccination as the non-healthcare workers. [27] interviewed nurses in Turkey about Covid-19related stress at work. They established that high stress at work is mirrored by high stress at home and with family. The diminution of stress and anxiety that accompanies a better knowledge of the disease is a reflection of the 'rational understanding' leading to a decrease in stress, identified among nurses in Wuhan [51].

Emotional response of anxiety felt by participants about lack of cohesive hospital-wide strategy is reflected in data from Spanish Emergency Departments: [52] investigated opinions of new nurses dealing with the initial outbreak with similar themes arising, with one nurse stating '*Every day, there are changes and it's a mess. You'll come to work and say, "Where should I go? It is a little bit chaotic what's going on right now due to the current pandemic.'* The rapidly changing procedures and protocols for positive patients in Bermuda is a microcosm of the global situation- everything changes so quickly, so response must be rapid, pragmatic and decisive. Clear communication on the particulars of the changing protocol has proved to be a challenge for healthcare infrastructure in Bermuda, as the rest of the world. The evidence that Bermuda has fared relatively well in dealing with Covid-19 cases and vaccination rollout [53] suggests that government and healthcare services did follow an effective plan that has, to date, been relatively successful. If it were possible to keep frontline staff reassured of a plan, and its details, this may have gone some way to put nurses' minds at rest.

Emotional response of loneliness and isolation felt by nurses in Bermuda, due to stigmatisation by interactions with neighbours, friends, other members of the community at the, is detected elsewhere globally [54]. Feelings of uncertainty, isolation and depression are on the increase for many groups [1]. Stressful events at work and feelings of being stigmatised within the community mean that nurses are particularly at risk. There are channels for help and counselling within the hospital; for example the EAP counselling programme mentioned by P5; although they could be under-utilised. The strong connection that nurses felt amongst their colleagues is evidence of how important workplace bonds have become to nurses in Bermuda, for support and a listening ear after stressful events or a difficult shift.

Many guest workers nurses in Bermuda live alone; so do not have the benefit of loved ones nearby, exacerbating loneliness. This sentiment echoes the concerns of nurses interviewed in Wuhan, who expressed the need to see their loved ones and enjoy supportive relationships, which is crucial for dealing with stressful situations in a healthy way [1]. The other resource valued by participants, holistic activities such as spending time outside, and specifically exercises; was limited by the closure of the onsite hospital-subsidised gym. The Staff Wellness programme that encouraged staff who would otherwise not be interested in engaging in exercise was halted, due to concern first about Covid-19 transmission, then potentially needing to use the gymnasium as a staff testing facility. This prolonged closure was a barrier to staff getting exercise that they valued as an effective way to manage their stress and anxiety.

## 5.1 Strengths and limitations of the study

The study was limited in scope and sample size, with only interviewed 9 of 374 registered working at KEMH. This represents a snapshot of the participants alone and therefore cannot

be generalised. While data saturation suggests that views may be commonly held among nurses working at the facility; the results are not transferable. Some of the participants are known personally to the researcher. While this allowed for free and frank speaking; with participants stating their feelings open and honestly; the Hawthorne Effect could play a part in the data gathered. The researcher is, in addition, an inexperienced interviewer. Some research bias is detectable; which could have skewed results. The evidence that only 2 of 9 participants stated they had not been vaccinated is not a reflection of low vaccination rates of nurses in Bermuda at time of interviews [55]. It is conceivable that nurses who felt that their views may not have been popular; or were controversial; were less motivated to take part; this potentially increases sampling bias; and limits the generalisability of the findings.

## 6. Conclusions

The data gathered in this study illuminates the emotional responses of nurses in Bermuda. Initial fear of the virus was overcome by knowledge gained due to emerging research and, and nurses were reassured in their work due to the newly developed vaccine. Anxieties about the correct PPE and its availability; and lack of communication of a cohesive plan to floor staff exacerbate stress responses. Additionally, emotional response of loneliness; feelings of isolation caused by stigma; and barriers to spending time with loved ones contribute to a dissatisfaction at an unfulfilling social and home life. Nurses found solace in support from their colleagues; both inside and outside of work, enjoying the support found in strong bonds between co-workers, with increased cooperation and emotional support. Participants were given succour in time spent with loved ones and knowing they are safe, whilst enjoying holistic activities, to soothe workplace borne stresses.

It is in the interest of the population of Bermuda; as an isolated country, with a disproportionately elderly, unwell population that nurse' concerns are addressed in order to provide for their needs, and ultimately retain the workforce. The main limitation of the study is the small sample size; with limited generalisability; but as the first study of its kind in Bermuda, results are useful as an exploratory case study, with a view to further research into the views of a wider sample. The emotional response of nurses to workplace-based advertising for the Covid-19 vaccine, and associated uptake, will be a particular area for further research to investigate, as a matter of clinical urgency in the ongoing pandemic response. Some potential workplace policy interventions stem from the study, with the aim of improving the chances of positive emotional response of nurses during the pandemic:

## 7. Recommendations

At national level, the development of policy to implement support systems and relevant budgeting and service allocation to provide support for nurses during and after pandemics. This could include wellness initiatives, with subsidised gym and exercise memberships; increased availability of optional one-to-one counselling services. Nurses may utilise these services; and find them useful, if they had knowledge about them. Low cost measures such as adverts on individual institutions' intranet, or posters in staff rooms could educate nurses about counselling services, for example.

Further research is necessary in order to inform the development of region-specific tailored pandemic preparedness plan and training for nurses. A plan must be established for rollout of pandemic services to aid understanding of decision making and educate floor-level staff about what to anticipate. Education; drills and assessment could provide for smoother transition to pandemic-forward services, to assuage stress level. Feedback must be utilised from recipients to ensure relevance and practicality. This is a challenge due to the quickly evolving situation,

yet facilitating new protocol being cascaded to staff in handover huddles/team meetings could be a quick and effective way to keep staff up-to-date about changes, without requiring any further human or financial resources.

Education for all healthcare professionals on the utility of the vaccine as safe is essential, in order that they can make the informed decision to take the vaccine in order to protect themselves and their loved ones; and encourage others to do the same. Further research into the above recommendations would be necessary to gauge the impact on satisfaction; and retention of nurses in the field, and at institutional level

## Supporting information

**S1 File. PRISMA chart of scopus search strategy.**
(TIF)

**S2 File. Gantt chart of research schedule.**
(TIF)

**S3 File. Graph showing Covid-19 cases in Bermuda.**
(TIF)

**S4 File. Recruitment poster for research study.**
(TIF)

**S5 File. Bermuda hospitals board diversity nursing report.**
(TIF)

**S1 Appendix.**
(DOCX)

## Acknowledgments

The author wishes to thank Navami Leena for indispensable guidance throughout the planning, conducting and writing of the thesis. Also, Sammy Bishop, for her assistance with proofreading, formatting, and super human patience. Thanks also to everyone who participated in interviews, and the wider team at King Edward Memorial Hospital.

## Author Contributions

**Conceptualization:** Adam Moore.

**Data curation:** Adam Moore.

**Formal analysis:** Adam Moore.

**Investigation:** Adam Moore.

**Methodology:** Adam Moore.

**Project administration:** Adam Moore.

**Resources:** Adam Moore.

**Supervision:** Navami Leena.

**Visualization:** Adam Moore.

**Writing – original draft:** Adam Moore.

**Writing – review & editing:** Adam Moore.

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
