## [Decision Letter · Decision Letter 0]

27 Jan 2023

PONE-D-22-34212An Exploration of the Emotional Response among Nurses in Bermuda, during the Covid-19 PandemicPLOS ONE

Dear Dr. Moore,

Thank you for submitting your manuscript to PLOS ONE. After careful consideration, we feel that it has merit but does not fully meet PLOS ONE’s publication criteria as it currently stands. Therefore, we invite you to submit a revised version of the manuscript that addresses the points raised during the review process.

We look forward to receiving your revised manuscript.

Kind regards,

Fatma Refaat Ahmed, Ph.D.

Academic Editor

PLOS ONE

Journal Requirements:

Additional Editor Comments:

Abstract needs restructure to be more clear to the reader.

The study design is not clear

The study question should be more focused and specific. 

Qualitative data analysis should be provided.

Reviewers' comments:

Reviewer's Responses to Questions

**Comments to the Author**

1. Is the manuscript technically sound, and do the data support the conclusions?

Reviewer #1: Partly

Reviewer #2: Yes

2. Has the statistical analysis been performed appropriately and rigorously? 

Reviewer #1: Yes

Reviewer #2: N/A

3. Have the authors made all data underlying the findings in their manuscript fully available?

Reviewer #1: Yes

Reviewer #2: Yes

4. Is the manuscript presented in an intelligible fashion and written in standard English?

Reviewer #1: Yes

Reviewer #2: Yes

5. Review Comments to the Author

Reviewer #1: Respected Authors,

asafter warm greetings,

Kindly find the attached reviewed article for considereing the recommended changes for research paper quality improvement. there is required clarifications in the abstract and the methodology sections. the ethics approval is not clear in relation to the approving body as well as the validity of the interview qualitative questions is not clearly described . also, there is overlaping between two different research designs that need to be fixed.so working on the modifications will improve the quality of the papaer.

Regards,

Reviewer #2: Many thanks for your valuable work.

Summary of the Study

This was an explorative study that aims to explore the emotional responses of nurses who treat Covid-19 patients in Bermuda, with the objective of gaining a better understanding of their experience. They interviewed 9 nurses to get in-depth analysis and exploration of these emerging phenomena. They made also an extensive literature review about previous researches conducted during COVID-19 pandemic. Six themes were elicited and explored to shed the light on these phenomena at Bermuda. They found that there is an initial fear of virus that was overcomed by knowledge. Anxieties about the correct PPE and its availability; and lack of communication of a cohesive plan to frontline staff exacerbate stress responses. Additionally, emotional response of loneliness; feelings of isolation caused by stigma; and barriers to spending time with loved ones contribute to a dissatisfaction at an unfulfilling social and home life. Nurses found solace in support from their colleagues; both inside and outside of work, enjoying the support found in strong bonds between co-workers, with increased cooperation and emotional support. Participants were given succor in time spent with loved ones and knowing they are safe, whilst enjoying holistic activities, to soothe workplace borne stresses.

Strength Points

Study topic is interesting for the international reader; it is about very important phenomena that whole world was encountered and still complaining from its outcomes. The whole article is perfectly written. I feel easy to understand their work, the article was very consistent with itself. Finally, the study has a good implication for the clinical practice, education, and research.

Weak Points

None

6. PLOS authors have the option to publish the peer review history of their article (what does this mean?). If published, this will include your full peer review and any attached files.

Reviewer #1: **Yes: **Dr. Sally Mohammed Farghaly

Reviewer #2: No

---

## [Author Response · Author response to Decision Letter 0]

22 Feb 2023

Firstly, thanks much for reviewing this paper. I appreciate your expertise and patience. Below I listed changes to the paper as suggested, responding to your comments. Please do let me know if you have any further feedback. This is the first paper I´ve written for publication, so any guidance at all is much appreciated. I have edited the study design to be more clear, and the objectives to be more focused. Please see below for specific changes. Aside from Edits to language, a couple of missed spacebar hits, and a capital letter here and there, I have updated the pre-2016 references with more recent, relevant studies. References can be found at the bottom of this document. 

Thank again. 

Adam

(Page Number listed correspond to manuscript draft originally submitted) 

p. 8 ABSTRACT- fully rewritten with aim, methods, conclusions and recommendations, as suggested. I am aware that my abstract is quite lengthy in comparison to most research paper, but felt it was pertinent to address the findings of the literature review as well as the primary research conducted. 

p. 14 LITERATURE REVIEW and OBJECTIVES- I have delineated between the 2 sections of the paper clearly to make it more clear. I outline that a lit review will be conducted to get a picture of the international situation, and then primary research to assess the experience of nurses in Bermuda. As such, there are a few structural changes that I have made throughout the paper. Objective questions have been separated as suggested.

p.16 Edited by ensure that the reader understands this is a lit review of qualitative research papers, followed by conduction of qualitative research 

P20. Methodology- distinguish between lit review and quality research DONE

p. 21 Emotional response changed to ´Psychological response´ and both terms are clarified with reference to Chew (2020) and (Kählke et al., 2019)

p23. ETHICS: Clarified motivations for ethical research, and type of ethical approvals granted. I also edited to add a brief discussion of measures to ensure validity, reliability, with reference to relevant literature. 

New references:

Chew, N. W., Lee, G. K., Tan, B. Y., Jing, M., Goh, Y., Ngiam, N. J., ... & Sharma, V. K. (2020). A multinational, multicentre study on the psychological outcomes and associated physical symptoms amongst healthcare workers during COVID-19 outbreak. Brain, behavior, and immunity, 88, 559-565. [Accessed 06/02/2023]

Cypress, Brigitte S. EdD, RN, CCRN. Rigor or Reliability and Validity in Qualitative Research: Perspectives, Strategies, Reconceptualization, and Recommendations. Dimensions of Critical Care Nursing 36(4):p 253-263, 7/8 2017. | DOI: 10.1097/DCC.0000000000000253 [Acccessed 09/02/2023]

Hackett, A., & Strickland, K. (2018). Using the framework approach to analyse qualitative data: a worked example. Nurse researcher, 26(2). DOI:10.7748/nr.2018.e1580 [Accessed 06/02/23]

Kählke, F., Berger, T., Schulz, A., Baumeister, H., Berking, M., Cuijpers, P., ... & Ebert, D. D. (2019). Efficacy and cost-effectiveness of an unguided, internet-based self-help intervention for social anxiety disorder in university students: protocol of a randomized controlled trial. BMC psychiatry, 19(1), 1-12. [Accessed 10/02/2023]

Khosravani, M., Abedi, H., Rafiei, F., & Rahzani, K. (2017). The association between conscience understanding and clinical performance among nurses working at education hospital of Arak. Annals of Tropical Medicine and Public Health, 10(6). [Accessed 10/02/2023]

Lak, S., Zahedi, S., Davodabady, F., & Khosravani, M. (2018). Conscience understanding among nurses working at education hospital of Arak. Revista Latinoamericana de Hipertensión, 13(3), 246-250. [Accessed 10/02/2023]

Rose, Jeff & Johnson, C.W (2020) Contextualizing reliability and validity in qualitative research: toward more rigorous and trustworthy qualitative social science in leisure research, Journal of Leisure Research, 51:4, 432-451, DOI: 10.1080/00222216.2020.1722042 [Acccessed 09/02/2023]

---

## [Editor Report · Decision Letter 1]

6 Mar 2023

PONE-D-22-34212R1An Exploration of the Emotional Response among Nurses in Bermuda, during the Covid-19 PandemicPLOS ONE

Dear Dr. Moore,

Thank you for submitting your manuscript to PLOS ONE. After careful consideration, we feel that it has merit but does not fully meet PLOS ONE’s publication criteria as it currently stands. Therefore, we invite you to submit a revised version of the manuscript that addresses the points raised during the review process.

We look forward to receiving your revised manuscript.

Kind regards,

Fatma Refaat Ahmed, Ph.D.

Academic Editor

PLOS ONE

---

## [Author Response · Author response to Decision Letter 1]

28 Mar 2023

NB. Thanks for feedback Re Figure one. I have edited Manuscript, ensuring that I refer to Fig 1 in text. Changes visible on manuscript with track changes. Thanks. Adam

---

## [Decision Letter · Decision Letter 2]

4 Oct 2023

An Exploration of the Emotional Response among Nurses in Bermuda, during the Covid-19 Pandemic

PONE-D-22-34212R2

Dear Dr. Moore,

We’re pleased to inform you that your manuscript has been judged scientifically suitable for publication and will be formally accepted for publication once it meets all outstanding technical requirements.

Kind regards,

Fatma Refaat Ahmed, Ph.D.

Academic Editor

PLOS ONE

Additional Editor Comments (optional):

Reviewers' comments:

Reviewer's Responses to Questions

**Comments to the Author**

1. If the authors have adequately addressed your comments raised in a previous round of review and you feel that this manuscript is now acceptable for publication, you may indicate that here to bypass the “Comments to the Author” section, enter your conflict of interest statement in the “Confidential to Editor” section, and submit your "Accept" recommendation.

Reviewer #3: All comments have been addressed

Reviewer #4: (No Response)

2. Is the manuscript technically sound, and do the data support the conclusions?

Reviewer #3: Yes

Reviewer #4: Yes

3. Has the statistical analysis been performed appropriately and rigorously? 

Reviewer #3: N/A

Reviewer #4: N/A

4. Have the authors made all data underlying the findings in their manuscript fully available?

Reviewer #3: Yes

Reviewer #4: Yes

5. Is the manuscript presented in an intelligible fashion and written in standard English?

Reviewer #3: Yes

Reviewer #4: Yes

6. Review Comments to the Author

Reviewer #3: This is an excellent manuscript. Authors have responded It has a technically sound piece of scientific research with information that supports the conclusions.

I recommend to accept this manuscript for publication.

Reviewer #4: Great and informative Work!!!

Generally:

I request the author to read through one more time and pick interest in spacing some words that are combined.

Specifically:

1. Kindly request the author to correct the typographical error in the abstract under methodology, second sentence referring to subjects, the author wrote "suhjects"

2. Under introduction, second sentence, author is requested to used the past tense i.e. "The WHO states that as of ... there were ..."

3. Let the author review the chart/graph numbering Vs reference within the body e.g. graph 1 in the body is labelled as 1.2.1

7. PLOS authors have the option to publish the peer review history of their article (what does this mean?). If published, this will include your full peer review and any attached files.

Reviewer #3: **Yes: **Muhammad Arsyad Subu

Reviewer #4: **Yes: **Allan G. Nsubuga

---

## [Editor Report · Acceptance letter]

7 Nov 2023

PONE-D-22-34212R2 

An Exploration of the Emotional Response among Nurses in Bermuda, during the Covid-19 Pandemic 

Dear Dr. Moore:

I'm pleased to inform you that your manuscript has been deemed suitable for publication in PLOS ONE. Congratulations! Your manuscript is now with our production department. 

Kind regards, 

on behalf of

Dr. Fatma Refaat Ahmed 

Academic Editor

PLOS ONE